# Patterns of social-affective responses to trauma exposure and their relation to psychopathology

Sarah Thomas[1], Judith Schäfer[1], Philipp Kanske[1,2], Sebastian Trautmann[1,3,4]*

1 Institute of Clinical Psychology and Psychotherapy, Faculty of Psychology, Technische Universität Dresden, Dresden, Germany, 2 Max Planck Institute for Human Cognitive and Brain Sciences, Leipzig, Germany, 3 Department of Psychology, Faculty of Human Science, Medical School Hamburg, Hamburg, Germany, 4 ICPP Institute of Clinical Psychology and Psychotherapy, Medical School Hamburg, Hamburg, Germany

* sebastian.trautmann@medicalschool-hamburg.de

## Abstract

### Introduction

Traumatic event exposure is a risk factor for the development and maintenance of psychopathology. Social-affective responses to trauma exposure (e.g. shame, guilt, revenge, social alienation) could moderate this relationship, but little is known about their relevance for different types of psychopathology. Moreover, the interplay of different social-affective responses to trauma exposure in predicting psychopathology is poorly understood.

### Methods

In a sample of $N = 1321$ trauma-exposed German soldiers, we examined cross-sectional associations of trauma-related social alienation, revenge, guilt and shame with depressive disorder, alcohol use disorder, posttraumatic stress disorder and dimensional measures of depression and anxiety. Latent class analysis was conducted to identify possible patterns of social-affective responses to trauma exposure, and their relation to psychopathology.

### Results

All social-affective responses to trauma exposure predicted current posttraumatic stress disorder, depressive disorder, alcohol use disorder and higher depressive and anxiety symptoms. Three latent classes fitted the data best, reflecting groups with (1) low, (2) moderate and (3) high risk for social-affective responses to trauma exposure. The low-risk group demonstrated the lowest expressions on all psychopathology measures.

### Conclusions

Trauma-related social alienation, shame, guilt, and revenge are characteristic of individuals with posttraumatic stress disorder, depressive disorder, alcohol use disorder, and with higher anxiety and depressive symptoms. There was little evidence for distinctive patterns of social-affective responses to trauma exposure despite variation in the overall proneness

**Data Availability Statement:** All relevant data are within the supporting information files.

**Funding:** The present study was funded by the German Ministry of Defence (https://www.bmvg.de/de; grant number: E/U2AD/HD008/CF550,

awarded to Sebastian Trautmann and Hans-Ulrich Wittchen) and was based on a larger former original study funded by the German Ministry of Defence (https://www.bmvg.de/de; grant number: M/SAB X/9A004, awarded to Hans-Ulrich Wittchen, Sabine Schönfeld and Clemens Kirschbaum). The funders had no role in study design, data collection and analysis, decision to publish, or preparation of the manuscript.

**Competing interests:** The authors have declared that no competing interests exist.

to show social-affective responses. Social-affective responses to trauma exposure could represent promising treatment targets for both cognitive and emotion-focused interventions.

## Introduction

Exposure to traumatic events is an important risk factor for the development and maintenance of mental disorders [1]. Apart from posttraumatic stress disorder (PTSD), trauma exposure is particularly associated with the development of depressive disorder (DD) and alcohol use disorder (AUD) [2]. However, individuals vary considerably in their response to trauma exposure and the majority of individuals adjust well to the experience of severe stressful or traumatic events [3]. Numerous factors have been suggested to moderate the association between trauma exposure and psychopathology [4]. Social factors, which have received less attention so far, are among those variables that could have a decisive influence on mental health after trauma exposure [5]. On the one hand, social factors include reactions from the social environment, such as social acknowledgement and provided social support. On the other hand, social factors include reactions and perceptions of trauma-exposed individuals themselves, such as trauma disclosure, perceived social support and social-affective responses to trauma exposure (e.g. trauma-related shame and guilt) [5, 6].

Among social factors, social-affective responses to trauma exposure could be of particular importance. Following the socio-interpersonal model of PTSD by Maercker and Horn [6], social-affective responses to trauma exposure can be understood as complex mental states encompassing feelings, cognitions and motivations that relate to the social reality of an individual. Social-affective responses to trauma exposure can include positive responses such as compassion [7] but can also include negative responses, such as shame, guilt, revenge and social alienation [6, 8]. In line with the socio-interpersonal model of PTSD, most authors conceptualize guilt [9], revenge [10], shame and social alienation [8] as complex states that are relevant from both a cognitive and an emotion-based perspective of posttraumatic processing. Cognitive models of posttraumatic stress assume that dysfunctional trauma appraisals lead to negative cognitive schemas about the self and the world and produce a sense of ongoing threat accompanied by diminished self-efficacy [11, 12]. In this context, trauma-related shame, guilt, and social alienation, for example, have been considered both as elements and consequences of negative cognitive schemas about the self and the world [11, 12]. From an emotion-based perspective, shame and guilt, and in some interpretations also feelings of estrangement and vengefulness [6], are conceptualized as social emotions [13]. Social emotions are regarded as "cognition-dependent" emotions that require mental representations of both oneself and others and work in the service of a social goal [14]. Recent theories and empirical findings increasingly emphasize the importance of distressing social emotions as possible responses to trauma exposure [13]. Previous findings suggest that negative social-affective responses to trauma exposure are particularly high after man-made trauma [15] involving direct contact with the perpetrator [16].

Importantly, negative social-affective responses to trauma exposure could be important for posttraumatic processing beyond general trauma-related emotional distress and negative cognitions. Social-affective responses to trauma exposure such as shame, guilt, or social alienation may be particularly difficult to manage because they can threaten a person's sense of self and social identity [17] and could seriously affect social relationships by preventing individuals from perceiving and using potential social resources such as social support or group

membership [18]. Moreover, there is evidence that social-affective responses to trauma exposure such as shame keep individuals from seeking professional help [19]. In line with these assumptions, negative social-affective responses to trauma exposure have been associated with higher levels of psychopathology in previous studies [5]. Trauma-related guilt and shame have been investigated most frequently and are associated with higher levels of PTSD symptoms [20, 21], with some authors suggesting a model of guilt and shame-based PTSD [17]. Trauma-related guilt and shame are highly interrelated, but it is assumed that after trauma exposure the relationship between guilt and PTSD is more variable and less strong than the relationship between shame and PTSD [20–22]. Besides trauma-related shame and guilt, trauma-related social alienation has shown to be an important mediator of the association between trauma exposure and PTSD symptoms [23]. Trauma-related revenge phenomena have received less attention so far, although trauma-related revenge feelings and cognitions have found to be predictive of higher severity and maintenance of PTSD symptoms [10, 24]. To date, social-affective responses to trauma exposure have mainly been investigated with respect to PTSD. In addition, a few studies investigated the relationship of social-affective responses to trauma exposure with depressive symptoms [23, 25, 26], with anxiety symptoms [25] and with alcohol use [27]. In these studies, trauma-related shame and guilt have been associated with higher levels of depressive/anxiety symptoms [25], and trauma-related guilt has been associated with higher depressive symptoms [26] as well as with increased alcohol use. Moreover, there is evidence that trauma-related social alienation mediates the association between traumatic event exposure and depressive symptoms [23].

Taken together, negative social-affective responses to trauma exposure have been associated with higher levels of subsequent psychopathology. Previous studies have focused primarily on PTSD and less is known about associations with other psychopathologies. In addition, most studies have examined trauma-related shame and guilt, while other possible social-affective responses to trauma exposure have received less attention. We hypothesized that trauma-related shame, guilt, revenge and social alienation are positively associated with the presence of PTSD, DD and AUD as well as with higher depressive and anxiety symptoms. Based on previous studies indicating that, for instance, trauma-related shame is more relevant to PTSD than trauma-related guilt [22], we assumed that the analyzed social-affective responses to trauma exposure could be of varying importance for the investigated outcomes. However, as there are few studies on this to date, the present study represents an exploratory investigation of the strength of the associations between social-affective responses to trauma exposure (shame, guilt, revenge, social alienation) and categorical (DD, AUD, PTSD) as well as dimensional (depression, anxiety) measures of psychopathology.

Moreover, the interplay of different social-affective responses to trauma exposure in predicting mental health has rarely been studied. Thus, little is known about whether there could be distinct patterns of different social-affective responses to trauma exposure and whether they relate differentially to psychopathology. Therefore, in addition to examining individual associations, the second aim of the present study was to investigate whether there are distinguishable patterns of social-affective responses to trauma exposure and, if so, how these patterns relate differentially to categorical and dimensional measures of psychopathology.

## Materials and methods

### Participants and procedure

Data were collected between 27.04.2010 and 10.12.2010 as part of the cross-sectional component of a larger original study program [28] investigating mental health and its determinants in German military personnel. A comprehensive description of the design of the original study

can be found elsewhere [28]. The present study is a secondary analysis of data collected as part of this original study. A total of $N$ = 2372 German soldiers were included in the original study. To be eligible for inclusion in the original study, soldiers had to be at least 18 years old. For the purpose of the present study, only participants who had been exposed to at least one lifetime traumatic event according to the DSM-IV-TR A1 criterion [29] were included ($N$ = 1636). Since the low proportion of females in the German military would not have permitted adequate subgroup analysis, female soldiers ($n$ = 104) were excluded in the present study. Moreover, participants who had any missing values on the items measuring trauma-related shame ($n$ = 207), trauma-related guilt ($n$ = 206), trauma-related revenge ($n$ = 206) and trauma-related social alienation ($n$ = 204) were excluded. For the present study, this resulted in an analysis sample of $N$ = 1321 individuals. To ensure that there was no selective non-response in the sense that more distressed individuals did not respond to the items, we examined whether the participants excluded due to missing values ($N$ = 211) and the analysis sample ($N$ = 1321) differed with respect to the outcomes examined. There were no differences regarding the severity of depressive and anxiety symptoms and regarding the percentage of PTSD and AUD, but excluded individuals had a lower percentage of DD than included individuals (S1 Table). Fig 1 shows a flow chart of the study group.

Participation in the study was voluntary and confidential. Trained clinical psychologists completed informed consent procedures and conducted the assessments. Informed written consent was obtained from all participants. The core assessment instrument was the computer-assisted version of the Munich-Composite International Diagnostic Interview (DIA-X/M-CIDI) [30]. The instrument was complemented with the assessment of military-specific information and with supplementary questionnaires that allowed for the assessment of dimensional symptom severity. The study was approved by the Ethics Board of Technische Universität Dresden (EK 72022010).

## Measures

**Lifetime traumatic event exposure.** In the present study, to align with the DSM-5 [31], a traumatic event was defined according to DSM-IV-TR A1 criterion [29]. The presence and number of lifetime traumatic events was assessed with the military version of the Munich-Composite International Diagnostic Interview (DIA-X/M-CIDI) [30]. As part of the interview, participants were provided with a list of traumatic events [32] which had been enlarged to also include military-specific events [28].

**Social-affective responses to trauma exposure (past four weeks).** Items measuring social-affective responses to trauma exposure originated from a 73-item a priori version of the Posttraumatic Cognitions Inventory (PTCI) [11] that has been used previously [33]. Of those 73 items, 26 items were included in the original study [28] to measure different negative cognitive-affective reactions to trauma exposure, including *perceived permanent change, alienation from self and others, self-blame, preoccupation with unfairness and negative interpretations of symptoms*. From those items, those that described the social-affective responses to trauma exposure that were of interest for the present research question (trauma-related shame, guilt, revenge, social alienation) were selected for the present study. Current social-affective responses to trauma exposure (in the past four weeks) were assessed with respect to the worst traumatic event. Items were rated on a 5-point scale ("Strongly disagree", "rather disagree", "neutral", "rather agree", "strongly agree"). Since several response categories had too low counts to treat the variables as dimensional, they were operationalized as dichotomous variables (present vs. not present). As shown in the online supplement (S2 Table) only a very small percentage of participants agreed to the items. Given the male military sample, it is possible

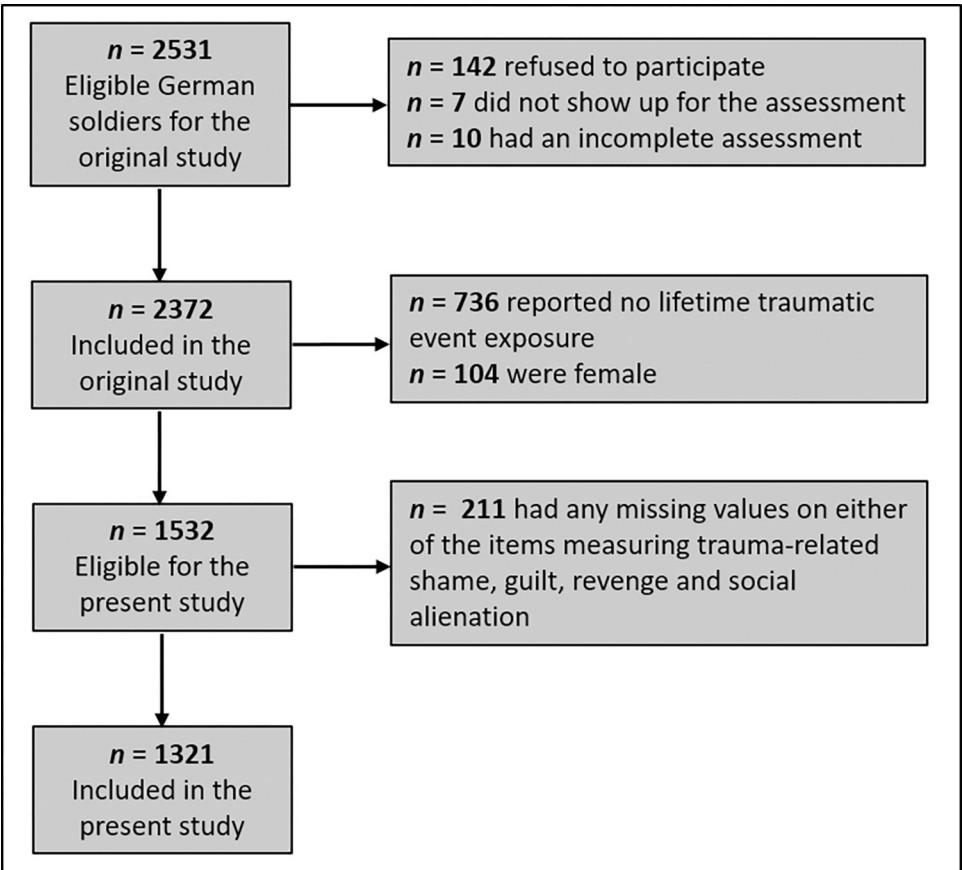

**Fig 1. Flow chart of the study group.**

that emotional and potentially stigmatizing constructs such as trauma-related shame, guilt, revenge, and social alienation were underreported [34]. Therefore, the middle response ("neutral"), which can be conceptualized as transition point between disagreement and agreement in Likert-type scales, was chosen as a cut-off for the presence of the respective social-affective response.

Guilt was defined as feelings and thoughts about having violated personal norms of right and wrong and being responsible for this wrongdoing (i.e. perceived lack of a justification for one's actions) [17]. Trauma-related guilt was rated as present if the item "The way I thought/ felt and behaved during the event is unforgivable" was not negated. Shame (external) relates to the experience of a negative social presentation and is characterized by feelings and thoughts of being devalued in the eyes of others and being looked down upon [17]. We decided to focus on external shame, since external shame has shown tighter links to psychopathology than internal shame [35] and could be easier to distinguish from guilt, as both guilt and internal shame refer to a negative self-evaluation, whereas external shame refers to the perception of being negatively evaluated by others [17]. External trauma-related shame was assessed with two items to be able to consider shame as a response to the actual presence of others during the traumatic event ("I embarrassed myself during the event") and as a response to the theoretical presence and judgment of others ("If people knew what happened, they would look down on me"). External trauma-related shame was rated as present if either of those two items was not negated (i.e. answered with "neutral", "rather agree" or "strongly agree"). We defined

revenge as the motivation to retaliate that results from feelings and thoughts of having been hurt wrongfully [36]. Trauma-related revenge was rated as present if the item "I want to punish the people who did this to me" was not negated. Social alienation was defined as feelings and thoughts of being disconnected from others [37]. Trauma-related social alienation was also measured with two items to consider both alienation in close relationships ("I will never be able to be close to other people again") as well as more generalized appraisals of disconnected-ness ("Other people do not understand me"). As for trauma-related shame, trauma-related social alienation was rated as present if either of those two items was not negated (i.e. answered with "neutral", "rather agree" or "strongly agree"). For trauma-related shame (0.94) and trauma-related social alienation (0.95) tetrachoric correlations between the items were high enough to allow the combination of the items into one construct.

**12-month mental disorders.** The prevalence of a DSM-IV-TR [29] diagnosis of DD, PTSD or AUD in the past 12 months was assessed using the military version of the Munich-Composite International Diagnostic Interview (DIA-X/M-CIDI [30]). The DIA-X/M-CIDI is a fully-standardized interview that allows a reliable [38] and valid [39] assessment of mental disorders for lifetime and in the past 12 months according to DSM-IV-TR [29] diagnostic criteria. DD was defined as the presence of either major DD or dysthymia in the past 12 months. To align with the DSM-5 [31], which collapsed abuse and dependence into a single disorder, AUD included those individuals who had met the criteria of either alcohol abuse or alcohol dependence in the past 12 months.

**Anxiety and depressive symptoms (past seven days).** Since it was deemed important to consider dimensional measures of psychopathology in addition to the categorical assessment of mental disorders [40], current anxiety and depressive symptoms (past seven days) were assessed with the German version of the Hospital Anxiety and DD Scale (HADS-D) [41]. The anxiety and the depression scale of the HADS-D each consist of seven items that are rated on a four-point scale. The response scales are anchored differently for each item and measure either the frequency or severity of symptoms or the severity of behavioral changes. A total sum score was calculated for anxiety symptoms (theoretical range 0–21) and for depressive symptoms (theoretical range 0–21). In the present sample, internal consistency was $\alpha = 0.75$ for the anxiety scale and $\alpha = 0.77$ for the depression scale.

## Data analysis

All analyses were performed with Stata 15.1 [42]. First, logistic regressions were calculated to examine whether and how strongly each individual social-affective response to trauma exposure (shame, guilt, revenge and social alienation) predicted the presence of DD, PTSD and AUD, respectively. In order to better assess the specificity of the individual associations, for each logistic regression, an additional model was calculated, adjusting for the respective comorbid disorders of DD, PTSD or AUD. Second, to complement the analyses by dimensional symptom measures, linear regressions were performed to examine individual associations of trauma-related shame, guilt, revenge and social alienation with depressive and anxiety symptoms. Again, models were re-calculated adjusting for anxiety symptoms in models with depressive symptoms as dependent variable, and vice versa.

Subsequently, Latent Class Analysis was performed to identify potential latent classes of patterns of social-affective responses to trauma exposure. The number of latent classes of social-affective responses to trauma exposure was determined using the Bayesian Information Criteria and Akaike Information Criteria. In a second step, subjects were assigned to a given latent class based on their posterior latent class membership probabilities. To examine whether latent classes of social-affective responses to trauma exposure were predictive of categorical

and/or dimensional measures of psychopathology, logistic and linear regressions were calculated with mental disorders and dimensional symptom measures as dependent and assigned latent class membership as predictor variable. Models were re-calculated adjusting for anxiety symptoms in models with depressive symptoms as dependent variable, and vice versa. Associations with diagnosis of PTSD, AUD or DD as dependent variable were adjusted for the respective comorbid disorders (PTSD, AUD, DD).

## Results

### Sample characteristics

Participants were male and had a mean age of 28.8 years ($SD$ = 7.6). Mean length of service was 8.5 years (SD = 7.6). The mean number of experienced traumatic events was 2.6 ($SD$ = 1.9). 39.7% of participants reported a directly experienced traumatic event related to combat or warzone experiences. 34.1% of participants reported a directly experienced traumatic event involving physical or sexual assault or abuse. 33.5% of participants reported a directly experienced traumatic event involving an accident, disaster or life threatening illness. 78.1% of participants reported a warzone or non-warzone related witnessed traumatic event (e.g. witness in the event of death, seeing a dead body or a seriously injured person).

There were 32.0% of participants who had children and 27.8% were married. Among the participants, 18.8% had a low educational level (9[th] grade), 63.2% had a middle (10[th] grade) educational level and 18.0% had a high (high school or higher) educational level. Of the participants, 1.7% rated their economic situation as "bad" or "very bad", 19.8% rated their economic situation to be at least sufficient and 78.5% rated their economic situation as "good" or "very good". Tetrachoric correlations between trauma-related shame, guilt, social alienation and revenge are presented in Table 1. High correlations were found between all social-affective responses to trauma exposure with the strongest correlation being between trauma-related guilt and shame (Rho = 0.88). The frequency of the presence of trauma-related revenge, social alienation, shame and guilt in the total sample and among individuals meeting criteria for PTSD, DD or AUD is shown in Table 2.

### Association of social-affective responses to trauma exposure with mental disorders and with dimensional symptom measures (anxiety and depression)

Table 3 shows the associations of trauma-related shame, guilt, revenge and social alienation with DD, PTSD and AUD. All associations were statistically significant. The strongest associations existed with respect to PTSD and with respect to trauma-related social alienation. The highest ORs were observed for associations between trauma-related social alienation and PTSD ($OR$ = 6.04, 95% CI = [3.47, 10.53], $p$ < .001) and between trauma-related social alienation and DD ($OR$ = 4.19, 95% CI = [2.39, 7.35], $p$ < .001). High ORs were also found for the

**Table 1.  Tetrachoric correlations between social-affective responses to trauma exposure.**

|  | Social alienation | Revenge | Shame | Guilt |
|---|---|---|---|---|
| Social alienation | 1 |  |  |  |
| Revenge | 0.77*** | 1 |  |  |
| Shame | 0.83*** | 0.84*** | 1 |  |
| Guilt | 0.82*** | 0.80*** | 0.88*** | 1 |

*** p < .001

**Table 2. Frequency of trauma-related social alienation, revenge, shame and guilt in individuals with a 12-month diagnosis of PTSD, DD and AUD.**

|  | Total sample N = 1321 | DD N = 53 | PTSD N = 54 | AUD N = 66 |
|---|---|---|---|---|
|  | n (%) | n (%) | n (%) | n (%) |
| Social alienation | 233 (17.6%) | 24 (45.3%) | 29 (53.7%) | 20 (30.3%) |
| Revenge | 231 (17.5%) | 15 (28.3%) | 23 (42.6%) | 25 (37.9%) |
| Shame | 155 (11.7%) | 14 (26.4%) | 18 (33.3%) | 16 (24.2%) |
| Guilt | 149 (11.3%) | 15 (28.3%) | 12 (22.2%) | 14 (21.2%) |

*Note*. DD = depressive disorder. PTSD = posttraumatic stress disorder. AUD = alcohol use disorder.

association between trauma-related shame and PTSD (*OR* = 4.12, 95% CI = [2.28, 7.46], *p* < .001), trauma-related revenge and PTSD (*OR* = 3.78, 95% CI = [2.16, 6.61], *p* < .001), and between trauma-related guilt and DD (*OR* = 3.34, 95% CI = [1.79, 6.23], *p* < .001). All associations were reduced when adjusted for comorbid disorders (Table 3) and there were no statistically significant associations any more between trauma-related revenge and DD and trauma-related guilt and PTSD.

Table 4 displays the associations between trauma-related shame, guilt, revenge and social alienation and anxiety and depressive symptoms. All associations were statistically significant. As for associations with mental disorders, the highest associations were observed with regard to trauma-related social alienation. Trauma-related social alienation predicted higher anxiety (β = 2.02, 95% CI = [1.64, 2.40]), *p* < .001) as well as higher depressive symptoms (β = 1.84, 95% CI = [1.46, 2.21], *p* < .001). A strong association was also found between trauma-related shame and depressive symptoms (β = 1.60, 95% CI = [1.15, 2.05], *p* < .001). All associations were reduced when adjusted for anxiety and depressive symptoms, respectively (Table 4). The association between trauma-related guilt and depressive symptoms was not statistically significant any more when adjusted for anxiety symptoms. When adjusted for depressive symptoms, there was no longer a significant association between trauma-related shame and anxiety symptoms.

**Table 3. Associations of trauma-related shame, guilt, revenge and social alienation with DD, PTSD and AUD.**

|  | DD |  |  | PTSD |  |  | AUD |  |  |
|---|---|---|---|---|---|---|---|---|---|
|  | OR | p | 95%CI | OR | p | 95%CI | OR | p | 95%CI |
| **Social alienation** |  |  |  |  |  |  |  |  |  |
| Unadjusted model | 4.19 | < .001 | [2.39, 7.35] | 6.04 | < .001 | [3.47, 10.53] | 2.13 | .007 | [1.23, 3.67] |
| Adjusted model | 3.31 | < .001 | [1.83, 5.98] | 5.06 | < .001 | [2.86, 8.96] | 1.80 | .044 | [1.01, 3.21] |
| **Revenge** |  |  |  |  |  |  |  |  |  |
| Unadjusted model | 1.92 | .037 | [1.04, 3.56] | 3.78 | < .001 | [2.16, 6.61] | 3.11 | < .001 | [1.85, 5.22] |
| Adjusted model | 1.45 | .269 | [0.75, 2.78] | 3.33 | < .001 | [1.87, 5.93] | 2.85 | < .001 | [1.68, 4.83] |
| **Shame** |  |  |  |  |  |  |  |  |  |
| Unadjusted model | 2.87 | .001 | [1.52, 5.42] | 4.12 | < .001 | [2.28, 7.46] | 2.57 | .002 | [1.42, 4.63] |
| Adjusted model | 2.20 | .021 | [1.13, 4.30] | 3.46 | < .001 | [1.88, 6.39] | 2.25 | .009 | [1.22, 4.13] |
| **Guilt** |  |  |  |  |  |  |  |  |  |
| Unadjusted model | 3.34 | < .001 | [1.79, 6.23] | 2.36 | .012 | [1.21, 4.59] | 2.23 | .011 | [1.21, 4.14] |
| Adjusted model | 2.90 | .001 | [1.52, 5.52] | 1.82 | .094 | [0.90, 3.66] | 1.99 | .033 | [1.06, 3.75] |

*Note*. DD = depressive disorder. PTSD = posttraumatic stress disorder. AUD = alcohol use disorder. Adjusted model: adjusted for the respective comorbid disorders of DD, PTSD or AUD.

**Table 4. Associations of trauma-related shame, guilt, revenge and social alienation with depressive and anxiety symptoms.**

| | Depressive Symptoms | | | Anxiety Symptoms | | |
|---|---|---|---|---|---|---|
| | β | p | 95%CI | β | p | 95%CI |
| **Social alienation** | | | | | | |
| Unadjusted model | 1.84 | < .001 | [1.46, 2.21] | 2.02 | < .001 | [1.64, 2.40] |
| Adjusted model | 0.59 | < .001 | [0.28, 0.89] | 0.86 | < .001 | [0.55, 1.16] |
| **Revenge** | | | | | | |
| Unadjusted model | 1.09 | < .001 | [0.70, 1.47] | 1.10 | < .001 | [0.71, 1.49] |
| Adjusted model | 0.39 | .010 | [0.09, 0.69] | 0.39 | .012 | [0.09, 0.69] |
| **Shame** | | | | | | |
| Unadjusted model | 1.60 | < .001 | [1.15, 2.05] | 1.33 | < .001 | [0.87, 1.79] |
| Adjusted model | 0.77 | < .001 | [0.42, 1.12] | 0.27 | .135 | [-0.09, 0.64] |
| **Guilt** | | | | | | |
| Unadjusted model | 1.08 | < .001 | [0.62, 1.54] | 1.35 | < .001 | [0.88, 1.82] |
| Adjusted model | 0.22 | .234 | [-0.14, 0.58] | 0.65 | < .001 | [0.29, 1.01] |

*Note.* Adjusted model: adjusted for depressive symptoms respectively anxiety symptoms.

## Latent class analysis

The fit statistics for different latent class solutions are displayed in Table 5. The model that fitted the data best was the one assuming three latent classes of social-affective responses to trauma exposure. The three latent classes model did not differ from a saturated model ($\chi^2(1) = 2.036$, $p = 0.154$). The frequencies of trauma-related shame, guilt, social alienation and revenge within each of the three latent classes of social-affective responses to trauma exposure are shown in Fig 2. The majority of individuals (79.2%) were assigned to a *low-risk group* for social-affective responses, 180 participants (13.6%) were assigned a *moderate-risk group* for social-affective responses and 95 participants (7.2%) to a *high-risk group* for social-affective responses to trauma exposure. The low-risk group was characterized by no or very low frequencies of social-affective responses to trauma exposure. Individuals in this group reported no trauma-related shame and no trauma-related social alienation, and only 6.7% of individuals reported trauma-related revenge and 2.2% reported trauma-related guilt. In the high-risk group, all individuals confirmed the presence of trauma-related shame, guilt and revenge and 92.6% confirmed the presence of trauma-related social alienation. In the moderate-risk group the percentage of individuals reporting trauma-related guilt (17.2%), shame (33.3%) and revenge (36.7%) was rather low, but a majority (80.6%) reported trauma-related social alienation.

**Table 5. Results of latent class analysis.**

| Model | AIC | BIC |
|---|---|---|
| One latent class | 4349.548 | 4370.293 |
| Two latent classes | 3278.232 | 3324.908 |
| Three latent classes | 3231.407 | 3304.013 |
| Four latent classes | 3235.372 | 3323.536 |

*Note.* AIC = Akaike's information criterion. BIC = Bayesian information criterion.

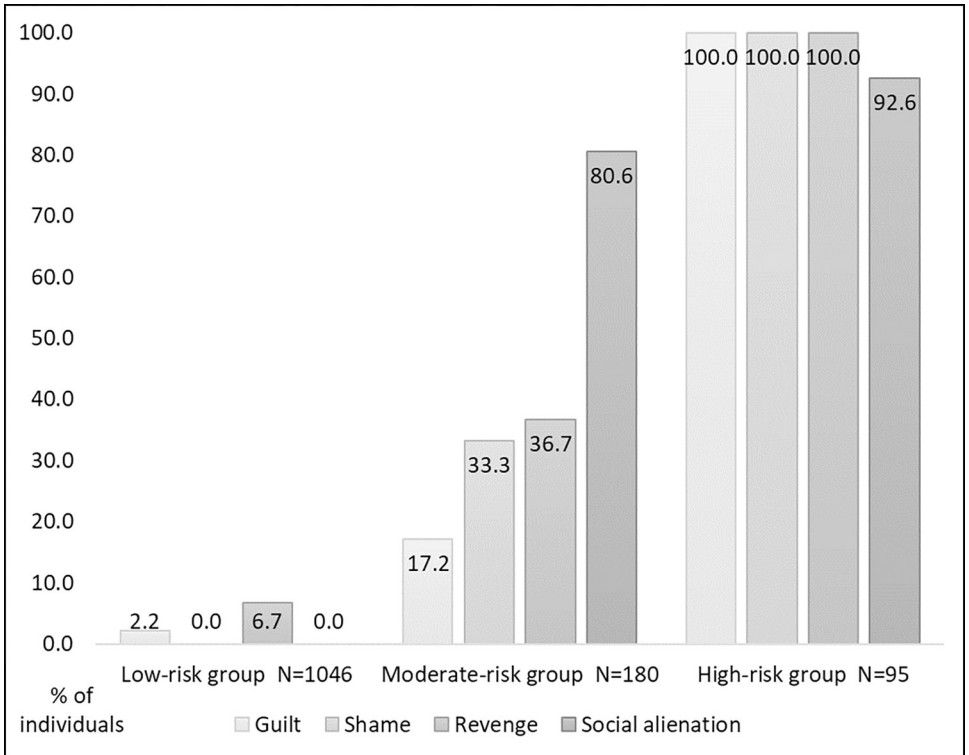

**Fig 2. Percentage of individuals reporting the presence of trauma-related guilt, shame, revenge and social alienation within each latent class of social-affective responses to trauma exposure.**

## Associations of latent class membership with mental disorders and with dimensional symptom measures (anxiety and depression)

Percentages of DD, PTSD, and AUD within the three latent classes of social-affective responses to trauma exposure are shown in Table 6. Descriptively, the highest percentage of PTSD (13.9%) and DD (11.1%) was in the moderate-risk group for social-affective responses to trauma exposure, followed by the high-risk group (PTSD: 6.3%, DD: 6.3%) and the low-risk group for social-affective responses to trauma exposure (PTSD: 2.2%, DD: 2.6%). In line with this, when compared to the low-risk group, the moderate-risk and the high-risk group for social-affective responses to trauma exposure had a higher risk for PTSD (Moderate vs. Low: $OR = 7.17$, 95% CI = [3.97, 12.95], $p < .001$; High vs. Low: $OR = 3.00$, 95% CI = [1.19, 7.56], $p = .020$) and for DD (Moderate vs. Low: $OR = 4.72$, 95% CI = [2.58, 8.61], $p < .001$; High vs. Low: $OR = 2.54$, 95% CI = [1.02, 6.33], $p = .044$). There were no statistical differences between the moderate-risk and the high-risk group in the percentage of PTSD and DD (Table 6).

With regard to the percentage of AUD, a slightly different pattern emerged: descriptively, the high-risk group for social-affective responses to trauma exposure had the highest percentage of AUD (9.5%), followed by the moderate-risk group (8.9%) and the low-risk group (3.9%). In line with this, the high-risk group ($OR = 2.57$ ,95% CI = [1.21, 5.45], $p = .014$) and the moderate-risk group for social-affective responses to trauma exposure ($OR = 2.39$, 95% CI = [1.31, 4.36], $p = .004$) had a higher risk for AUD than the low-risk group. The high-risk group and the moderate-risk group did not differ from each other with respect to the percentage of AUD (Table 6). Adjusting for comorbid disorders did not considerably change the described pattern of results (Table 6).

**Table 6. Percentage of DD, PTSD and AUD within each latent class of social-affective responses to trauma exposure and associations between latent class membership and diagnoses.**

|  | Low-risk (N = 1046) | Moderate-risk (N = 180) | High-risk (N = 95) | Moderate-risk vs. Low-risk | High-risk vs. Low-risk | High-risk vs. Moderate-risk |
|---|---|---|---|---|---|---|
|  | % | % | % | OR (95%CI) | OR (95%CI) | OR (95%CI) |
| **DD** | 2.6 | 11.1 | 6.3 |  |  |  |
| Unadjusted model |  |  |  | 4.72***(2.58, 8.61) | 2.54*(1.02, 6.33) | 0.54(0.21, 1.39) |
| Adjusted model |  |  |  | 3.62***(1.91, 6.86) | 2.24(0.89, 5.64) | 0.62(0.23, 1.63) |
| **PTSD** | 2.2 | 13.9 | 6.3 |  |  |  |
| Unadjusted model |  |  |  | 7.17***(3.97, 12.95) | 3.00*(1.19, 7.56) | 0.42(0.17, 1.06) |
| Adjusted model |  |  |  | 5.92***(3.22, 10.88) | 2.62*(1.02, 6.69) | 0.44(0.17, 1.14) |
| **AUD** | 3.9 | 8.9 | 9.5 |  |  |  |
| Unadjusted model |  |  |  | 2.39*(1.31, 4.36) | 2.57*(1.21, 5.45) | 1.07(0.46, 2.53) |
| Adjusted model |  |  |  | 2.01*(1.07, 3.79) | 2.39*(1.12, 5.12) | 1.19(0.50, 2.83) |

*Note*. DD = depressive disorder. PTSD = posttraumatic stress disorder. AUD = alcohol use disorder. Adjusted model: adjusted for the respective comorbid disorders of DD, PTSD or AUD.

* p < .05

** p < .01

*** p < .001

Dimensional measures of anxiety and depressive symptoms for each latent class of social-affective responses to trauma exposure are presented in Table 7. Similar to what was found for DD and for PTSD, the moderate-risk group descriptively had the highest mean values for depressive symptoms ($M$ = 3.9) and for anxiety symptoms ($M$ = 4.8), followed by the high-risk group and the low-risk group (Table 7). In accordance with this, the moderate-risk group ($\beta$ = 2.09, 95% CI = [1.67, 2.52], $p$ < .001) and the high-risk group ($\beta$ = 1.36, 95% CI = [0.80, 1.93], $p$ < .001) had higher anxiety symptoms than the low-risk group for social-affective responses to trauma exposure. Moreover, the high-risk group for social-affective responses to trauma

**Table 7. Dimensional symptom measures of anxiety and depression in each latent class of social-affective responses to trauma exposure and associations between latent class membership and dimensional symptom measures.**

|  | Low-risk (N = 1046) | Moderate-risk (N = 180) | High-risk (N = 95) | Moderate-risk vs. Low-risk | High-risk vs. Low-risk | High-risk vs. Moderate-risk |
|---|---|---|---|---|---|---|
|  | M (SD) | M (SD) | M (SD) | β (95%CI) | β (95%CI) | β (95% CI) |
| **Depressive symptoms** | 1.9 (2.4) | 3.9 (3.3) | 3.4 (3.4) |  |  |  |
| Unadjusted model |  |  |  | 1.97***(1.55, 2.38) | 1.51***(0.95, 2.06) | -0.46(-1.11, 0.20) |
| Adjusted model |  |  |  | 0.68***(0.34, 1.02) | 0.67**(0.23, 1.11) | -0.01(-0.52, 0.50) |
| **Anxiety symptoms** | 2.7 (2.5) | 4.8 (3.3) | 4.1 (3.5) |  |  |  |
| Unadjusted model |  |  |  | 2.09***(1.67, 2.52) | 1.36***(0.80, 1.93) | -0.73*(-1.40, -0.06) |
| Adjusted model |  |  |  | 0.84***(0.50, 1.19) | 0.40(-0.04, 0.85) | -0.44(-0.96, 0.08) |

*Note*. M = Mean value. SD = Standard Deviation. Adjusted model: adjusted for depressive symptoms respectively anxiety symptoms.

* p < .05

** p < .01

** p < .001

exposure had lower anxiety symptoms than the moderate-risk group (β = -0.73, 95% CI = [-1.40, -0.06], *p* = .032).

The moderate-risk group (β = 1.97, 95% CI = [1.55, 2.38], *p* < .001) and the high-risk group (β = 1.51, 95% CI = [0.95, 2.06], *p* < .001) also had higher depressive symptoms than the low-risk group. The moderate-risk and the high-risk group did not differ with respect to the magnitude of depressive symptoms (Table 7).

When adjusted for anxiety respectively depressive symptoms, all associations were reduced (Table 7), and the high-risk group did no longer differ from the low-risk and the moderate-risk group with respect to anxiety symptoms.

## Discussion

The first aim of the present study was to examine individual associations of social-affective responses to trauma exposure (revenge, social alienation, guilt, shame) with categorical (PTSD, DD; AUD) and dimensional (anxiety, depression) measures of psychopathology. The second aim was to investigate potential latent classes of patterns of social-affective responses to trauma exposure and their relation to categorical and dimensional measures of psychopathology.

All social-affective responses to trauma exposure were related to a higher risk for all examined mental disorders (PTSD, DD, AUD) as well as to higher levels of depressive and anxiety symptoms. Interestingly, for both DD and PTSD, as well as for depressive and for anxiety symptoms, the highest point estimates of associations were observed with trauma-related social alienation. So far, trauma-related social alienation has received relatively little attention. A meta-analysis from 2020 found only nine studies that investigated associations between trauma-related alienation and PTSD symptoms, but suggested a large effect size [37]. Among those nine studies, two studies compared trauma-related fear, anger, betrayal, shame, self-blame and alienation with respect to different psychological symptoms [23, 43]. One study found that, when investigated together, only alienation predicted PTSD and depressive symptoms [23] and the other study demonstrated that trauma-related alienation was the only variable that predicted all forms of investigated trauma-related distress (PTSD, dissociation, and depression symptoms) across different samples [43].

In the present study, the strong association between trauma-related social alienation and PTSD might partly be explained to the fact that trauma-related social alienation overlaps with the DSM-IV-TR PTSD criterion "feeling of detachment or estrangement from others" [29]. However, it seems unlikely that the association was attributable to this overlap alone, as trauma-related social alienation also most strongly predicted DD, anxiety symptoms and depressive symptoms. Trauma-related social alienation could contribute to psychopathology as it could interfere with an individual's sense of (social) identity, foster insecure attachment styles and associated emotional distress [23, 43] and lead to a reduced capacity to benefit from potential social resources [18]. However, a relationship in the opposite causal direction seems also conceivable, since individuals with a psychopathology of depression, anxiety or posttraumatic stress often suffer from diminished interest or pleasure, demonstrate avoidance behavior and experience stigma, which could all lead to social withdrawal and promote feelings and cognitions of social alienation. This could result in a vicious cycle in which social alienation fosters psychopathology and higher psychopathology in turn reinforces social alienation.

Besides trauma-related social alienation, trauma-related shame was the strongest predictor of PTSD, whereas trauma-related guilt was the weakest predictor of PTSD. This is in line with previous studies demonstrating that after trauma exposure shame is more strongly related to PTSD than guilt [20–22]. Shame might be more aversive than guilt, because it does not only

refer to one's perceived misbehavior in a specific situation (e.g. "I did something bad"), but to more global negative self-appraisals (e.g. "I am bad") as well as to the perception of being devalued in the eyes of others [21]. In line with this, affective models of posttraumatic stress assume that trauma-related shame contributes substantially to the development and maintenance of trauma-associated disorders [44]. Shame stimulates self-protective impulses, possible leading to hypervigilance and avoidance behavior in interpersonal situations or to social withdrawal [44]. In the long term, this could foster persistent negative beliefs about the self and the social environment, as no corrective experiences are made. Moreover, trauma-related shame could stimulate the suppression of trauma-related thoughts and memories and reluctance to talk about the traumatic event [44].

In the present study, trauma-related guilt appeared to be of particular relevance for DD, which may be partly due to the fact that excessive or inappropriate guilt is a potential symptom of major DD. Previous studies have shown that perceived lack of control during a traumatic event is positively associated with trauma-related guilt [45]. It has been suggested that trauma-related guilt could serve to avoid feelings of helplessness following the trauma, as guilt conveys a sense of control [46] Trauma-related guilt could contribute to psychopathology by preventing the processing of primary emotions during the trauma.

Trauma-related revenge was the strongest predictor of AUD. Contrary to trauma-related shame and guilt, revenge has received very little attention as a social-affective response to trauma exposure, although interpersonal aggression is common among trauma survivors [5]. Similar to trauma-related guilt, it has been suggested that trauma-related revenge could function as an emotion avoiding strategy that inhibits the processing of primary emotions during the trauma, such as helplessness. [10]. Our findings highlight the importance of identifying not only self-critical responses to trauma exposure (e.g., shame, guilt) but also hostile reactions towards others.

Besides investigating individual associations between social-affective responses to trauma exposure and psychopathology, the second aim of this study was to examine possible latent classes of social-affective responses to trauma exposure and their relation to psychopathology. Three latent classes of social-affective responses to trauma exposure were identified that fitted the data best reflecting groups with low, moderate and high risk for negative social-affective responses to trauma exposure. The found latent classes seem to primarily reflect the overall proneness to experience negative social-affective responses to trauma exposure. There appear to be few systematic patterns of social-affective responses to trauma exposure with a high risk for one social-affective response and a low risk for other social-affective responses to trauma exposure. Therefore, individuals who are more prone to self-critical social-affective responses to trauma exposure (e.g. guilt, shame) also seem to be more prone to report hostile reactions (e.g. revenge) and to report trauma-related social alienation. This is consistent, for example, with theories assuming that shame can result in externalization of blame and anger towards others as well as in social withdrawal [17]. It is also in line with theories suggesting that feelings and cognitions of revenge often activate shame and guilt [47].

In the present study, one exception was that in the moderate-risk group, trauma-related social alienation was reported with high likelihood, whereas the risk of reporting other social-affective responses to trauma exposure was considerably smaller. After trauma exposure, the threshold to experience trauma-related social alienation might therefore be relatively low. One might also speculate that reporting trauma-related social alienation is less stigmatized than reporting trauma-related revenge, guilt, or shame.

As could be expected, the low-risk group for social-affective responses to trauma exposure had the lowest risk for PTSD, AUD and DD and the lowest levels of depressive and anxiety symptoms. A more surprising finding was that the high-risk group did not show higher levels

of psychopathology than the moderate-risk group for social-affective responses to trauma exposure. In contrary, the high-risk group even had lower anxiety symptoms than the moderate-risk group. A possible explanation could be that the moderate-risk and the high-risk group differed not only in terms of the likelihood with which individuals in these groups reported social-affective responses to trauma exposure, but also in the way they coped with distressing feelings and thoughts. It is conceivable that some individuals in the moderate-risk group relied more heavily on avoidant coping strategies (e.g. rumination, experiential avoidance, thought suppression) to down-regulate the experience of negative social-affective responses to trauma exposure. Such avoidant strategies, however, are related to higher levels of internalizing and distress-related psychopathology, such as symptoms of PTSD, depression and anxiety [48, 49]. Another explanation could be that, in the present study, trauma-related social alienation was particularly relevant for psychopathology, and individuals in the moderate-risk and in the high-risk group differed little in the likelihood with which they reported trauma-related social alienation. Taken together, it appears necessary to consider not only the mere presence of social-affective responses to trauma exposure but also their regulation and other potentially relevant moderating factors to understand the relationship between social-affective responses to trauma exposure and psychopathology.

This study has several limitations. (1) We examined a relatively healthy sample with an average low frequency of self-reported negative social-affective responses to trauma exposure and low levels of psychopathology. This is a limitation in three regards. First, it reduces the variance in the variables under investigation, which could have led to an underestimation of group differences or associations. Second, it leads to limited generalizability to populations with higher levels of social-affective responses to trauma exposure and symptomatology. Third, social-affective responses to trauma exposure were operationalized as dichotomous variables due to their low variance, leading to a loss of information compared to a dimensional measure. (2) We examined a male, military sample, which limits the generalizability of the findings. (3) The possibility of underreporting of mental health problems in a male, military sample [34] could have been a potential source of measurement bias. Moreover, we used retrospective self-report instruments that can be subject to recall bias and to response bias, including neutral or extreme response bias. (4) The present study is secondary analysis of data originally collected in 2010. All hypotheses were therefore formulated post-hoc, which has to be considered when interpreting the findings. (5) Moreover, at the time of the original study in 2010 [28], diagnoses were based on DSM-IV-TR criteria [29], so that a transfer to DSM-5 disorders [31] is only possible to a limited extent. (6) There were no validated instruments available to assess all of the examined social-affective responses to trauma exposure. Despite careful theoretical considerations, the validity of the used items remains unclear. (7) This was a cross-sectional study, so no definite conclusions can be made about the temporal sequence of the variables studied. Furthermore, we cannot indicate the length of time between the worst traumatic event and the time of assessment in the original study, but it is likely that for some individuals, there were long time periods between exposure and assessment. This could have led to an underestimation of psychopathology or social-affective responses to trauma exposure, as these may have already been remitted before the study assessment. Longitudinal studies are needed to investigate the relationship between social-affective responses to trauma exposure and subsequent psychopathology.

## Conclusions

Despite the limitations described, several important implications can be drawn from the findings of the present study. Our results indicate that trauma-related social alienation, shame,

guilt, and revenge are likely phenomena in individuals who meet criteria for AUD, DD and PTSD as well as in individuals with higher levels of depressive and anxiety symptoms. This is important since previous research suggests that negative social-affective responses to trauma exposure contribute to a higher severity and to the maintenance of psychopathology [10, 19]. In addition, it has been demonstrated that trauma-related shame, guilt and alienation are associated with poorer outcomes in exposure based treatments [18, 50] and that within-person change in trauma-related shame and guilt predict changes in psychopathology during treatment [50]. This underlines the importance of considering social-affective responses to trauma exposure as possible treatment targets. More specifically, individuals experiencing negative social-affective responses to trauma exposure could particularly benefit from trauma-focused cognitive interventions that challenge dysfunctional trauma interpretations [18, 51]. Additionally, emotion-focused interventions aimed at promoting (self-)compassion represent a promising approach for individuals experiencing self-critical responses such as shame and guilt after trauma exposure [51] or hostile responses such as trauma-related revenge. Moreover, as compassion-focused interventions also aim to enforce social connectedness, they might be helpful for individuals experiencing trauma-related social alienation. Other emotion-focused interventions such as dialectic behavioral therapy [52] have also been shown to reduce trauma-related shame and guilt in PTSD [53]. In addition, emotion-focused interventions may be particularly helpful for trauma-exposed individuals if they exhibit high levels of experiential avoidance and/or impulsivity, both of which are common in AUD, for example [54, 55]. Finally, individuals experiencing trauma-related social alienation may benefit from interpersonal skills training alongside cognitive and emotion-focused methods.

Our findings further suggest that it is important for both researchers and clinicians to keep in mind that the presence of self-critical responses to trauma exposure (e.g. shame, guilt) is often accompanied by hostile responses (e.g. trauma-related revenge) and trauma-related social alienation. Similarly, individuals who present primarily with hostile responses towards others could at the same time have problems with reduced self-esteem [10] and may strongly experience trauma-related shame and guilt. Therefore, it seems important to also assess those social-affective responses to trauma exposure that may not be initially reported by patients, especially if these responses could be perceived as stigmatizing. For future studies, it would be a valuable aim to investigate whether trauma-related guilt, shame, revenge and social alienation could be used as possible indicators for the presence of mental disorders such as PTSD, DD and AUD.

To further understand the potential causal pathways between social-affective responses to trauma exposure and subsequent psychopathology, future studies should investigate the relationship between social-affective responses to trauma exposure and mental disorders in prospective longitudinal studies, ideally with multiple assessments shortly after trauma exposure. Upcoming studies should also examine the extent to which findings of the present study can be replicated in different samples, including different demographic groups (high-risk groups vs. general population), different trauma types, different gender groups, and groups with higher levels of psychopathology and negative social-affective responses to trauma exposure.

## Supporting information

**S1 Table. Comparison of participants included versus excluded due to missing data.**
(DOCX)

**S2 Table. Distribution of items measuring trauma-related guilt, revenge, shame and social alienation.**
(DOCX)

**S1 File.**
(CSV)

**S2 File.**
(XLSX)

## Acknowledgments

Sabine Schönfeld, Clemens Kirschbaum and Hans-Ulrich Wittchen contributed to the planning of the former original study. Beyond the co-authors (Sebastian Trautmann and Judith Schäfer), Christin Thurau, Michaela Galle, Kathleen Mark and Anke Schumann were involved in the logistical handling. Moreover, the staff of the "Centre for Psychiatry and Posttraumatic Stress" Berlin supported the fieldwork in the former original study.

## Author Contributions

**Conceptualization:** Sarah Thomas, Philipp Kanske, Sebastian Trautmann.

**Data curation:** Sarah Thomas, Judith Schäfer, Sebastian Trautmann.

**Formal analysis:** Sarah Thomas, Sebastian Trautmann.

**Funding acquisition:** Sebastian Trautmann.

**Investigation:** Judith Schäfer, Sebastian Trautmann.

**Methodology:** Sarah Thomas, Sebastian Trautmann.

**Project administration:** Sebastian Trautmann.

**Supervision:** Philipp Kanske, Sebastian Trautmann.

**Validation:** Sebastian Trautmann.

**Visualization:** Sarah Thomas.

**Writing – original draft:** Sarah Thomas.

**Writing – review & editing:** Sarah Thomas, Judith Schäfer, Philipp Kanske, Sebastian Trautmann.

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
