## [Decision Letter · Decision Letter 0]

15 Nov 2023

PONE-D-23-22194Patterns of social-affective responses to trauma exposure and their relation to psychopathologyPLOS ONE

Dear Dr. Trautmann,

Thank you for submitting your manuscript to PLOS ONE. After careful consideration, we feel that it has merit but does not fully meet PLOS ONE’s publication criteria as it currently stands. Therefore, we invite you to submit a revised version of the manuscript that addresses the points raised during the review process.

Dear Authors,

Please review the feedback given by the reviewers (especially reviewer 6, who recommended rejection) and revise the manuscript accordingly. 

Thanks

Lakshit

We look forward to receiving your revised manuscript.

Kind regards,

Lakshit Jain, MD

Academic Editor

PLOS ONE

Journal Requirements:

The study was logistically supported by the staff of the “Centre for Psychiatry and Posttraumatic Stress” in Berlin. Sabine Schönfeld, Clemens Kirschbaum and Hans-Ulrich Wittchen contributed to the planning of the former study program. Beyond the co-authors (Sebastian Trautmann and Judith Schäfer), Christin Thurau, Michaela Galle, Kathleen Mark and Anke Schumann were involved in the logistical handling.

This study was funded by the German Ministry of Defence (https://www.bmvg.de/de; grant number: E/U2AD/HD008/CF550, awarded to SeT). The funders had no role in study design, data collection and analysis, decision to publish, or preparation of the manuscript.

Reviewers' comments:

Reviewer's Responses to Questions

**Comments to the Author**

1. Is the manuscript technically sound, and do the data support the conclusions?

Reviewer #1: Yes

Reviewer #2: Yes

Reviewer #3: Yes

Reviewer #4: Partly

Reviewer #5: Yes

Reviewer #6: Partly

Reviewer #7: Yes

2. Has the statistical analysis been performed appropriately and rigorously? 

Reviewer #1: Yes

Reviewer #2: Yes

Reviewer #3: Yes

Reviewer #4: I Don't Know

Reviewer #5: I Don't Know

Reviewer #6: Yes

Reviewer #7: Yes

3. Have the authors made all data underlying the findings in their manuscript fully available?

Reviewer #1: No

Reviewer #2: No

Reviewer #3: No

Reviewer #4: No

Reviewer #5: Yes

Reviewer #6: No

Reviewer #7: Yes

4. Is the manuscript presented in an intelligible fashion and written in standard English?

Reviewer #1: Yes

Reviewer #2: Yes

Reviewer #3: Yes

Reviewer #4: Yes

Reviewer #5: Yes

Reviewer #6: Yes

Reviewer #7: Yes

5. Review Comments to the Author

Reviewer #1: Thank you for the opportunity to review this important piece of trauma exposure and its correlation to psychopathology. The paper adds to the present literature on this topic and my recommendation is to accept the paper with the following edits:

1. The authors can consider adding the social-affective responses as a factor in the social factors in line 50.

2. While defining the social-affective responses, the authors may want to indicate if these responses are among the people facing the trauma or among the people they interact with.

3. The authors have done an excellent job with the introduction, what would be interesting to know is is there any literature on the impact of the type of trauma (combat, sexual, accidental) on the social-affective responsivity. This may help identify if there are differences between the study group (military personnel) who are probably prone to combat trauma and other groups.

4. Given the date of data collection, this posits to a weakness of the relevance of the study to todays time given the use of DSM5-TR criteria at this time. The authors may want to comment on this.

5. In the measures section, the authors talk of lifetime exposure, while the scale measures last 4 weeks responses. This begs to question the temporal interpretations of the study. The authors may want to comment on this.

6. The authors may want to include a flowchart of the study group and exclusion criteria and how it led to the final n.

7. The authors may want to use the full terms for PTSD and Depressive Disorders prior to using acronyms.

The paper highlights an important correlation between social affective responsivity and mental health illness. The paper uses an appropriate methodology and rigid statistical analysis. However, the paper could ennumerate further about the importance of the risk stratification and its utility in treatment modalities and diagnostics for AUD.

Reviewer #2: Thank you for the opportunity to review this well written paper that looks at relationship between social affective responses to trauma exposure and psychopathology.

The cross sectional associations between shame, guilt, revenge and social alienation with both categorical disorders (depressive, anxiety, PTSD and alcohol use) and symptoms of anxiety/depression are examined.

Overall the manuscript was extremely well written with an easy to understand style.

The design of the study was simple and explained well. The methods section was written particularly well, with the tables and figure easy to read and interpret.

While previous studies have focused on PTSD, the inclusion of depression/anxiety and alcohol use disorders and the examination of interplay between each of these adds value to literature.

The authors do acknowledge the limitations of the study well including the homogeneity of the sample being mostly male, military with less dysfunction and likely to underreport.

-The data slows that social alienation is probably the most important factor for psychopathology after a traumatic event. This is a highlight of the results.

-The other interesting finding is that of the high risk group being less likely to be associated with psychopathology that the moderate risk group. The authors do try to explain various possible reasons for this. But this does pique interest and needs to be potentially explored in future studies.

I did have a couple of thoughts

-It seems like the authors used data from a previous study that was collected all the way back in 2010. If so then the authors need to acknowledge this as a secondary analysis of existing data and include it as a limitation. When conducting such a study, is there a hypothesis that the authors had prior to looking at the data ? This needs to be more clear in the manuscript.

-Also the data on social affective responses were collected from PTCI questionnaire. Where there other items on the questionnaire that were of interest or why were they not included. This could be explained a little bit more. Again if this a secondary look at existing data (which is fine for a study), a little bit more detail on where the data comes from would be helpful to provide context.

-Social affective réponses were measured in the ‘past four weeks’. Does that mean from the time of assessment or four weeks after traumatic exposure ? It its from the time of assessment then does amount of time from the actual traumatic event matter ? The authors should comment on this.

-In line 263, under latent class analysis section of the results, please revise to ‘all associations were statistically significant’

Overall good manuscript and useful study. Good work.

Reviewer #3: The research paper provides a detailed examination of the association between social-affective responses towards trauma and the subsequent manifestation of psychopathology. It posits that individuals reacting with feelings of alienation, urges for revenge, guilt, and shame to traumatic incidents are more likely to develop mental disorders, such as post-traumatic stress disorder (PTSD), depressive disorder (DD), alcohol use disorder (AUD), and display higher symptoms of depression and anxiety. The study utilized a sample of over two thousand German soldiers who experienced at least one traumatic event in their lifetime.

Feedback:

The argument that social-affective responses to trauma could predict the likelihood of psychopathology is adequately supported by the available data. Since the study is cross-sectional, it can only highlight correlations but cannot definitively establish causality. Despite such limitations, the sample size and statistical analysis techniques used strengthen the validity of the arguments presented. The paper's clarity regarding the methodology employed for data collection enhances the overall validity of the arguments concerning the prediction of psychopathology. The conclusion regarding the potential utility of social-affective responses as targets for treatment interventions is well-founded in the data. While the paper places substantial emphasis on the connection between social-affective responses and psychopathology, its argument would benefit from an exploration of potential mediators or variables that might influence this relationship. The paper conscientiously acknowledges the absence of longitudinal data and reliance on self-reported measures as limitations, which is a candid self-assessment that lends credibility to its arguments.

While the writing style retains a professional demeanor, the overall flow is interrupted by certain instances of colloquial language. Maintaining a professional tone could greatly improve the article. Example: Instead of using expressions like "It's a no-brainer," prefer a more formal approach like "It's clear…". Pay attention to the consistency and precision of language. It is noticed that the same concepts expressed in different parts of the paper are phrased variably, leading to reader confusion. Example: Once the term “latent classes of social-affective responses” has been introduced, it should be used in the same manner all the way through. The abstract gives a comprehensive insight into the study. However, its dense language and the lack of space between distinct points make it difficult for the reader to absorb the information. A short and crisp abstract in bullet points could be user-friendly. Using uniformity in abbreviations would be helpful. Certain abbreviations are introduced early on but are mysteriously dropped halfway through the text. It may confuse the reader and disrupt the text's fluidity. Consistent use of abbreviations after they are introduced ensures an easy read. Overall, the paper presents a well-investigated overview, but improvements in language and narrative style could significantly enhance its readability.

The article demonstrates commendable citation practices, showcasing a well-structured and consistent use of a recognized citation style. Each citation is complete, accurate, and directly supports the content. The inclusion of primary sources and the avoidance of over-citation contribute to the article's clarity and credibility. The use of up-to-date references further enhances the article's reliability.

The article is of high quality, with minor language revisions and adjustments in the use of abbreviations, as specified above and in accordance with the PLOS guidelines. It is also important to ensure data availability, as per these guidelines.

Reviewer #4: This is a well-written manuscript. I liked reading this manuscript and believe that it is very promising. At the same time, I identified couple of issues that require the authors’ attention.

The manuscript is based on impressive empirical evidence and makes an original contribution but there should be some comment on possible bias like reporting bias of the study participants.

Author should add that for the future studies, sample size should include different type of trauma victims (not just like military sample as in this study which is one of the limitations of this study as we can't generalized the result to different types of trauma exposure).

Author should also comment on inclusion and exclusion criteria and sample population that how many of them have already diagnosis of SUD before joining the military and hx of Trauma exposure other than combat related.

Also recommend the author to separate the discussion, result and conclusion section instead of everything under one section of discussion.

Reviewer #5: 1. 90 However, there is also some evidence regarding other forms of posttraumatic psychopathology, such as depressive symptoms. please give ref and elobrate this

2. Similar to PTSD, trauma-related shame (22) and guilt (22, 23) have been associated with higher levels of depressive symptoms. This is not clear, could you please rephrase this, and expand this.

3. There is reductant content from lines 90 to 95; please avoid reductacy.

4. Previous studies have focused primarily on PTSD and less is known about associations with other psychopathologies such as depressive disorder (DD) and AUD. What are those studies, please give ref as well as discuss them

5. authors used Social-affective responses by responses. There is high likley chances of response bias, how did you address: Participants may have a tendency to always select a certain response option, such as "neutral," without fully considering the item. This bias leads to a lack of variability in responses and may not accurately reflect individuals' true opinions.

6. Regarding assessing external shame: By including neutral responses as an indication of the presence of external shame, participants who may not actually experience external shame could be misclassified as experiencing it. This misclassification could skew the data and result in an overestimation of the prevalence or intensity of external shame in the population under study. Including only neutral responses to indicate the presence of external shame may also overlook individuals who truly experience external shame but choose not to respond neutrally to the items. This could lead to an underestimation of the prevalence or intensity of external shame in the sample. This is concering to me to assess this way, has anyone else assessed like this in previosuly published studies?

7. Aim within intro and discussion are somewhat different, let's say not the same

Discssion: Examining individual associations of social-affective responses (revenge, social alienation, guilt, shame) to trauma exposure with indicators of psychopathology. The aim is to explore the relationship between these social-affective responses and categorical and dimensional measures of mental disorders.

intro: focused on investigating associations of negative social-affective responses (social alienation, revenge, guilt, shame) to trauma exposure with specific mental disorders (DD, AUD, and PTSD), as well as with dimensional measures of depression and anxiety. The aim is also to examine if distinct patterns of trauma-related social-affective responses exist and how these patterns are differentially related to mental disorders and dimensional symptom measures.

Reviewer #6: The manuscript looks at an important topic, help-seeking in medical students, and the impact of educational climate, and stigma on help-sseking. However, the manuscript has several challenges.

1. It does not utilize an established framework for help-seeking, which is probably why key variables are missing (e.g., knowledge/literacy, attitudes, social support, severity of symptoms) that are part of help-seeking frameworks and models (e.g., based on the theory of planned behavior). Thus, it is difficult to connect the research to previous work in this area (which the authors do to a small extent). Since the authors are only presenting this data, it is not clear what information might be available in the larger project, so I have to assume that it is not possible to conduct a more thorough analysis that also connects more closely to previous research in the field and thus is innovative or promising in this regard.

2. Moreover, the concept of medical school factors also lacks a clear framework. Aspects like educational climate are self-reported perceptions by the students and thus not organizational factors. The items assess aspects like social relationships with other students. Since social isolation can be a key symptom of mental ill health, this assessment is highly confounded. If participants experience more severe mental health problems, this might lead them to socialize less and thus report a less friendly climate or less belongingness. Since severity of symptoms can also be associated with stigma, this would need to be addressed. However, the current assessment does not allow for any disentanglement of these effects and therefore the conclusions remain speculative.

3. To provide more impact to their work, the authors could include more organizational variables that are available to them, such as size of the classes/cohorts, and universities (and thus potential social networks), workload per year, degree of rurality of each institution etc. This would add a more nuanced organizational perspective here. Also, interaction effects between different levels could be tested (e.g., size and gender).

4. The recruitment and sampling needs to be expanded. How were students selected and approached? In what way were they representative of the student body? How was missing data accounted for? Was an attrition analysis performed? etc. This should be expanded.

Reviewer #7: The article is very beautifully written and talks about social affective responses to trauma. It was interesting to learn that social alienation was strongly associated with PTSD, DD and for depressive and anxiety symptoms. Also revenge was strongest predictor for AUD. These findings can be useful treatment targets for future treatment of these disorders.

2. The tables are self explanatory.

ONE MINOR MISTAKE: references 31( Line 583) Wittchen HU et al and Reference 43(Line 619) Kummerle S et al are not in english.

6. PLOS authors have the option to publish the peer review history of their article (what does this mean?). If published, this will include your full peer review and any attached files.

Reviewer #1: No

Reviewer #2: No

Reviewer #3: **Yes: **Aditi Sharma

Reviewer #4: No

Reviewer #5: No

Reviewer #6: **Yes: **Samuel Tomczyk

Reviewer #7: **Yes: **Jasleen Kaur

---

## [Author Response · Author response to Decision Letter 0]

23 Jan 2024

Response to Editor:

We have ensured that the manuscript meets PLOS ONE’s journal requirements, including those for file naming.

“The study was logistically supported by the staff of the “Centre for Psychiatry and Posttraumatic Stress” in Berlin. Sabine Schönfeld, Clemens Kirschbaum and Hans-Ulrich Wittchen contributed to the planning of the former study program. Beyond the co-authors (Sebastian Trautmann and Judith Schäfer), Christin Thurau, Michaela Galle, Kathleen Mark and Anke Schumann were involved in the logistical handling.”

“This study was funded by the German Ministry of Defence (https://www.bmvg.de/de; grant number: E/U2AD/HD008/CF550, awarded to SeT). The funders had no role in study design, data collection and analysis, decision to publish, or preparation of the manuscript.”

We have updated the funding statement so that the funding statement also includes funding received for the former original study.

We would like to update the funding statement as follows:

“The present study was funded by the German Ministry of Defence (https://www.bmvg.de/de; grant number: E/U2AD/HD008/CF550, awarded to Sebastian Trautmann and Hans-Ulrich Wittchen) and was based on a larger former original study funded by the German Ministry of Defence (https://www.bmvg.de/de; grant number: M/SAB X/9A004, awarded to Hans-Ulrich Wittchen, Sabine Schönfeld and Clemens Kirschbaum). The funders had no role in study design, data collection and analysis, decision to publish, or preparation of the manuscript.”

The amended funding statements are also included in the revised cover letter.

The acknowledgment section did not include funding related information. The “Centre for Psychiatry and Posttraumatic Stress” in Berlin was not a funder, but did only support the fieldwork. We have clarified this in the acknowledgment section and have modified the acknowledgement section as follows:

“Sabine Schönfeld, Clemens Kirschbaum and Hans-Ulrich Wittchen contributed to the planning of the former original study. Beyond the co-authors (Sebastian Trautmann and Judith Schäfer), Christin Thurau, Michaela Galle, Kathleen Mark and Anke Schumann were involved in the logistical handling. Moreover, the staff of the “Centre for Psychiatry and Posttraumatic Stress” Berlin supported the fieldwork in the former original study.”

We uploaded an anonymized data set that allows the replication of all analyses described in the manuscript. All relevant data are within the manuscript and its supporting information files. All demographic and military-related variables had to be removed since these represent sensitive data in this particular military sample.

The amended data availability statement has been included in the revised cover letter.

Response to Reviewers:

Reviewer #1:

Thank you for the opportunity to review this important piece of trauma exposure and its correlation to psychopathology. The paper adds to the present literature on this topic and my recommendation is to accept the paper with the following edits:

We thank the reviewer for the positive evaluation of our manuscript.

1. The authors can consider adding the social-affective responses as a factor in the social factors in line 50.

The social-affective responses have been added as a social factor (line 53-54).

2. While defining the social-affective responses, the authors may want to indicate if these responses are among the people facing the trauma or among the people they interact with.

We thank the reviewer for the remark. We have clarified that social-affective responses are among the people facing the trauma (line 52-53).

3. The authors have done an excellent job with the introduction, what would be interesting to know is is there any literature on the impact of the type of trauma (combat, sexual, accidental) on the social-affective responsivity. This may help identify if there are differences between the study group (military personnel) who are probably prone to combat trauma and other groups.

We thank the reviewer for the positive evaluation of the introduction. So far, there are only very few studies investigating the impact of the type of trauma on social-affective responses. These studies suggest that negative social-affective responses are highest in man-made trauma involving direct contact with the perpetrator. For instance, La Bash et al. (2014) demonstrated that individuals exposed to an interpersonal trauma (e.g. physical abuse, sexual abuse) reported higher levels of trauma-related shame than individuals exposed to an impersonal trauma (e.g. natural disaster). In addition, using a sample of veterans, Meade et. al (2022) found that trauma-related guilt prior to treatment was higher after sexual trauma than after combat trauma. We have added this information to the introduction (line 73-75).

La Bash H, Papa A. Shame and PTSD symptoms. Psychological Trauma: Theory, Research, Practice, and Policy. 2014;6(2):159-66.

Meade EA, Smith DL, Montes M, Norman SB, Held P. Changes in guilt cognitions in intensive PTSD treatment among veterans who experienced military sexual trauma or combat trauma. Journal of Anxiety Disorders. 2022;90:102606.

4. Given the date of data collection, this posits to a weakness of the relevance of the study to todays time given the use of DSM5-TR criteria at this time. The authors may want to comment on this.

We agree with the reviewer that the use of DSM-IV-TR criteria at the time of the original study represents a weakness with regard to the relevance of the present study. In the present study, however, we have already adopted the DSM-5 criteria in some respects. To determine the presence of a traumatic event, we used the DSM-IV-TR A1 criterion, which is consistent with the DSM-5, where criterion A2 (the person’s response involved intense fear, helplessness, or horror) was removed (line 158-159). In addition, alcohol use disorder was defined as the presence of either alcohol dependence or alcohol abuse, which is also consistent with DSM-5, which collapsed abuse and dependence into a single disorder (line 217-219). However, we are aware that there are important changes between DSM-IV-TR and DSM-5, particularly with respect to PTSD, which could not be taken into account in the present study. We have therefore included the use of DSM-IV-TR criteria as a limitation (line 529-532).

5. In the measures section, the authors talk of lifetime exposure, while the scale measures last 4 weeks responses. This begs to question the temporal interpretations of the study. The authors may want to comment on this.

We thank the reviewer for the remark. We have not described the measurement of all relevant constructs clearly enough in the methods section. Only traumatic event exposure was assessed with respect to lifetime exposure. All other constructs (mental disorders, symptoms, social-affective responses to trauma exposure) were recorded with respect to the time period specified in the respective section. The respective time periods are now more clearly described in the methods section (line 158,164,211,220). Moreover, we now describe the measurement of trauma exposure in the methods section in greater detail in a separate paragraph (line 158-163).

6. The authors may want to include a flowchart of the study group and exclusion criteria and how it led to the final n.

We have added a flow chart of the study group illustrating all exclusion criteria and how it let to final n (Figure 1).

7. The authors may want to use the full terms for PTSD and Depressive Disorders prior to using acronyms.

We are now using the full terms for PTSD and Depressive Disorders prior to using abbreviations (line 42-44).

The paper highlights an important correlation between social affective responsivity and mental health illness. The paper uses an appropriate methodology and rigid statistical analysis. However, the paper could ennumerate further about the importance of the risk stratification and its utility in treatment modalities and diagnostics for AUD.

We discuss implications for treatment (line 553-568) and diagnostics (line 575-578) in relation to all disorders and have added additional information to these sections. We have also added specific information on AUD (line 563-566).

Reviewer #2: 

Thank you for the opportunity to review this well written paper that looks at relationship between social affective responses to trauma exposure and psychopathology.

The cross sectional associations between shame, guilt, revenge and social alienation with both categorical disorders (depressive, anxiety, PTSD and alcohol use) and symptoms of anxiety/depression are examined.

Overall the manuscript was extremely well written with an easy to understand style.

The design of the study was simple and explained well. The methods section was written particularly well, with the tables and figure easy to read and interpret.

While previous studies have focused on PTSD, the inclusion of depression/anxiety and alcohol use disorders and the examination of interplay between each of these adds value to literature.

The authors do acknowledge the limitations of the study well including the homogeneity of the sample being mostly male, military with less dysfunction and likely to underreport.

-The data slows that social alienation is probably the most important factor for psychopathology after a traumatic event. This is a highlight of the results.

-The other interesting finding is that of the high risk group being less likely to be associated with psychopathology that the moderate risk group. The authors do try to explain various possible reasons for this. But this does pique interest and needs to be potentially explored in future studies.

We thank the reviewer for the positive feedback on the manuscript.

I did have a couple of thoughts

-It seems like the authors used data from a previous study that was collected all the way back in 2010. If so then the authors need to acknowledge this as a secondary analysis of existing data and include it as a limitation. When conducting such a study, is there a hypothesis that the authors had prior to looking at the data? This needs to be more clear in the manuscript.

We thank the reviewer for the remark. These aspects have not been described clearly enough in the article. We have clarified in the methods section that the present study is a secondary analysis of existing data (line 131-132) and have added information on the original study. We have also added as a limitation that the present study is a secondary analysis of existing data and that all hypotheses were therefore formulated post-hoc (line 527-529).

Based on the described literature, we hypothesized that the investigated predictors (trauma-related shame, guilt, revenge and social alienation) and the investigated outcomes (PTSD, DD, AUD, anxiety symptoms, depressive symptoms) were positively associated. Based on previous studies indicating that, for instance, trauma-related shame is more relevant to PTSD than trauma-related guilt, we assumed that the analyzed social-affective responses to trauma exposure could be of varying importance for the investigated outcomes. However, as there are very few studies to date that compare the relevance of different social-affective responses for different forms of psychopathology – or studies that investigate the interplay of social-affective responses in relation to psychopathology – we did not formulate specific hypotheses in this regard, but investigated these associations exploratively. We have now clarified this in the introduction (line 108-118´7).

-Also the data on social affective responses were collected from PTCI questionnaire. Where there other items on the questionnaire that were of interest or why were they not included. This could be explained a little bit more. Again if this a secondary look at existing data (which is fine for a study), a little bit more detail on where the data comes from would be helpful to provide context.

We thank the reviewer for the remark. We have now described the source and the selection of the items in greater detail and have provided further information about the original study (line 164-172).

Social affective réponses were measured in the ‘past four weeks’. Does that mean from the time of assessment or four weeks after traumatic exposure? It its from the time of assessment then does amount of time from the actual traumatic event matter? The authors should comment on this.

We have clarified and highlighted that „past four weeks“ refers to time of the assessment (line 164, 173).

We cannot precisely determine the time period between the worst traumatic event and the time of assessment. It is likely that for some individuals, there were long time periods between exposure and assessment. This could have led to an underestimation of psychopathology or social-affective responses to trauma exposure, as these may have already been remitted before the study assessment. We have included it as a limitation that we cannot indicate the length of time between the worst traumatic event and the time of assessment (line 535-540).

-In line 263, under latent class analysis section of the results, please revise to ‘all associations were statistically significant’

We thank the reviewer for noticing this error. This has been changed.

Overall good manuscript and useful study. Good work.

We are thankful for the overall positive evaluation of our manuscript.

Reviewer #3: 

The research paper provides a detailed examination of the association between social-affective responses towards trauma and the subsequent manifestation of psychopathology. It posits that individuals reacting with feelings of alienation, urges for revenge, guilt, and shame to traumatic incidents are more likely to develop mental disorders, such as post-traumatic stress disorder (PTSD), depressive disorder (DD), alcohol use disorder (AUD), and display higher symptoms of depression and anxiety. The study utilized a sample of over two thousand German soldiers who experienced at least one traumatic event in their lifetime.

Feedback:

1. The argument that social-affective responses to trauma could predict the likelihood of psychopathology is adequately supported by the available data. Since the study is cross-sectional, it can only highlight correlations but cannot definitively establish causality. Despite such limitations, the sample size and statistical analysis techniques used strengthen the validity of the arguments presented. The paper's clarity regarding the methodology employed for data collection enhances the overall validity of the arguments concerning the prediction of psychopathology. The conclusion regarding the potential utility of social-affective responses as targets for treatment interventions is well-founded in the data. While the paper places substantial emphasis on the connection between social-affective responses and psychopathology, its argument would benefit from an exploration of potential mediators or variables that might influence this relationship. The paper conscientiously acknowledges the absence of longitudinal data and reliance on self-reported measures as limitations, which is a candid self-assessment that lends credibility to its arguments.

We thank the reviewer for this feedback. Information on potential mediators between social-affective responses and psychopathology has been added. We are now discussing potential mediators of social alienation (line 432-443), shame (line 447-457), guilt (line 460-464) and revenge (line 468-470) in more detail.

2. While the writing style retains a professional demeanor, the overall flow is interrupted by certain instances of colloquial language. Maintaining a professional tone could greatly improve the article. Example: Instead of using expressions like "It's a no-brainer," prefer a more formal approach like "It's clear…". Pay attention to the consistency and precision of language. It is noticed that the same concepts expressed in different parts of the paper are phrased variably, leading to reader confusion. Example: Once the term “latent classes of social-affective responses” has been introduced, it should be used in the same manner all the way through. The abstract gives a comprehensive insight into the study. However, its dense language and the lack of space between distinct points make it difficult for the reader to absorb the information. A short and crisp abstract in bullet points could be user-friendly. Using uniformity in abbreviations would be helpful. Certain abbreviations are introduced early on but are mysteriously dropped halfway through the text. It may confuse the reader and disrupt the text's fluidity. Consistent use of abbreviations after they are introduced ensures an easy read. Overall, the paper presents a well-investigated overview, but improvements in language and narrative style could significantly enhance its readability.

We thank the reviewer. We have carefully reviewed the whole article to identify and revise any potential inadequate use of informal language. We have also assured uniformity in the introduction (line 42-44) and subsequent use of abbreviations. We have also ensured that the same concepts are expressed identically throughout the article. To this end, changes have been made throughout the article, which are highlighted accordingly. We have shortened the abstract.

3. The article demonstrates commendable citation practices, showcasing a well-structured and consistent use of a recognized citation style. Each citation is complete, accurate, and directly supports the content. The inclusion of primary sources and the avoidance of over-citation contribute to the article's clarity and credibility. The use of up-to-date references further enhances the article's reliability.

We thank the reviewer for the positive evaluation.

4. The article is of high quality, with minor language revisions and adjustments in the use of abbreviations, as specified above and in accordance with the PLOS guidelines. It is also important to ensure data availability, as per these guidelines.

We thank the reviewer for the positive evaluation of our manuscript.

We uploaded an anonymized data set that allows the replication of all analyses described in the manuscript. All relevant data are within the manuscript and its supporting information files. All demographic and military-related variables had to be removed since these represent sensitive data in this particular military sample.

Reviewer #4: 

This is a well-written manuscript. I liked reading this manuscript and believe that it is very promising. At the same time, I identified couple of issues that require the authors’ attention. The manuscript is based on impressive empirical evidence and makes an original contribution but there should be some comment on possible bias like reporting bias of the study participants.

We thank the reviewer for the remark. In the limitation section, we are addressing possible selection bias (line 515-524) and possible measurement/reporting bias, including the possibility of underreporting of mental health problems in a male, military sample (line 524-526) as well as possible recall bias and response bias in self-reporting instruments (line 526-527).

Author should add that for the future studies, sample size should include different type of trauma victims (not just like military sample as in this study which is one of the limitations of this study as we can't generalized the result to different types of trauma exposure).

We added that future studies should examine different trauma types (line 586).

Author should also comment on inclusion and exclusion criteria and sample population that how many of them have already diagnosis of SUD before joining the military and hx of Trauma exposure other than combat related.

We thank the reviewer for the remark. We have added information on inclusion criteria (133-134) and have added a flow chart of the study group that illustrates all exclusion criteria (Figure 1). Furthermore, we now describe the study group in more detail (line 255-262) and have added information on the type of traumatic events reported by participants (line 257-262). We have no information on how many soldiers had an AUD prior to military service. However, this is beyond the scope of this manuscript, as entry into the military was not relevant for the present research question, but the presence of a traumatic event was the relevant inclusion criterion for the present study.

Also recommend the author to separate the discussion, result and conclusion section instead of everything under one section of discussion.

Result, discussion and conclusion section have been separated.

Reviewer #5:

1. 90 However, there is also some evidence regarding other forms of posttraumatic psychopathology, such as depressive symptoms. please give ref and elobrate this

We thank the reviewer for the remark. Previous findings on other psychopathologies are now described more clearly and reference is given to each individual study (line 96-103).

2. Similar to PTSD, trauma-related shame (22) and guilt (22, 23) have been associated with higher levels of depressive symptoms. This is not clear, could you please rephrase this, and expand this.

The respective paragraph has been rephrased and expanded (line 96-103).

3. There is reductant content from lines 90 to 95; please avoid reductacy.

The respective paragraph has been rephrased to avoid redundant content.

4. Previous studies have focused primarily on PTSD and less is known about associations with other psychopathologies such as depressive disorder (DD) and AUD. What are those studies, please give ref as well as discuss them

The respective sentence has been rephrased (line 105-106). Previous studies on PTSD are cited and discussed from line 85-95. Previous studies investigating other psychopathologies (depressive symptoms, anxiety symptoms, alcohol use) are cited and discussed from line 96-103.

5. authors used Social-affective responses by responses. There is high likley chances of response bias, how did you address: Participants may have a tendency to always select a certain response option, such as "neutral," without fully considering the item. This bias leads to a lack of variability in responses and may not accurately reflect individuals' true opinions.

We have added it as limitation that we used self-reporting instruments that can be subject to recall bias and to response bias, including neutral or extreme response bias (line 526-527).

6. Regarding assessing external shame: By including neutral responses as an indication of the presence of external shame, participants who may not actually experience external shame could be misclassified as experiencing it. This misclassification could skew the data and result in an overestimation of the prevalence or intensity of external shame in the population under study. Including only neutral responses to indicate the presence of external shame may also overlook individuals who truly experience external shame but choose not to respond neutrally to the items. This could lead to an underestimation of the prevalence or intensity of external shame in the sample. This is concering to me to assess this way, has anyone else assessed like this in previosuly published studies?

We thank the reviewer for the remark. The measurement of trauma-related external shame has been misleadingly described by us and we have corrected this (line 197-199). Trauma-related external shame was rated as present if either of the two items assessing trauma-related external shame was not negated, i.e. answered with “neutral”, “rather agree” or “strongly agree”.

7. Aim within intro and discussion are somewhat different, let's say not the same

Discssion: Examining individual associations of social-affective responses (revenge, social alienation, guilt, shame) to trauma exposure with indicators of psychopathology. The aim is to explore the relationship between these social-affective responses and categorical and dimensional measures of mental disorders.

intro: focused on investigating associations of negative social-affective responses (social alienation, revenge, guilt, shame) to trauma exposure with specific mental disorders (DD, AUD, and PTSD), as well as with dimensional measures of depression and anxiety. The aim is also to examine if distinct patterns of trauma-related social-affective responses exist and how these patterns are differentially related to mental disorders and dimensional symptom measures.

We thank the reviewer for noticing this. The respective parts in the introduction (line 108-125) and in the discussion (line 410-415) have been modified. It has been clarified in the introduction and in the discussion that the two aims of the present study were 1) the investigation of individual associations between social-affective responses to trauma exposure and categorical (PTSD, AUD, DD) and dimensional (anxiety, depression) measures of psychopathology and 2) the investigation of possible patterns of social-affective responses to trauma exposure and their relation to categorical and dimensional measures of psychopathology

Reviewer #6:

The manuscript looks at an important topic, help-seeking in medical students, and the impact of educational climate, and stigma on help-sseking. However, the manuscript has several challenges.

Unfortunately, it seems that the comments of Reviewer 6 do not refer to our manuscript but to another manuscript (investigating help-seeking in medical students), therefore we could not respond to the comments of Reviewer 6.

1. It does not utilize an established framework for help-seeking, which is probably why key variables are missing (e.g., knowledge/literacy, attitudes, social support, severity of symptoms) that are part of help-seeking frameworks and models (e.g., based on the theory of planned behavior). Thus, it is difficult to connect the research to previous work in this area (which the authors do to a small extent). Since the authors are only presenting this data, it is not clear what information might be available in the larger project, so I have to assume that it is not possible to conduct a more thorough analysis that also connects more closely to previous research in the field and thus is innovative or promising in this regard.

2. Moreover, the concept of medical school factors also lacks a clear framework. Aspects like educational climate are self-reported perceptions by the students and thus not organizational factors. The items assess aspects like social relationships with other students. Since social isolation can be a key symptom of mental ill health, this assessment is highly confounded. If participants experience more severe mental health problems, this might lead them to socialize less and thus report a less friendly climate or less belongingness. Since severity of symptoms can also be associated with stigma, this would need to be addressed. However, the current assessment does not allow for any disentanglement of these effects and therefore the conclusions remain speculative.

3. To provide more impact to their work, the authors could include more organizational variables that are available to them, such as size of the classes/cohorts, and universities (and thus potential social networks), workload per year, degree of rurality of each institution etc. This would add a more nuanced organizational perspective here. Also, interaction effects between different levels could be tested (e.g., size and gender).

4. The recruitment and sampling needs to be expanded. How were students selected and approached? In what way were they representative of the student body? How was missing data accounted for? Was an attrition analysis performed? etc. This should be expanded.

Reviewer #7: 

The article is very beautifully written and talks about social affective responses to trauma. It was interesting to learn that social alienation was strongly associated with PTSD, DD and for depressive and anxiety symptoms. Also revenge was strongest predictor for AUD. These findings can be useful treatment targets for future treatment of these disorders. 2. The tables are self explanatory.

We thank the reviewer for the positive evaluation of our manuscript.

ONE MINOR MISTAKE: references 31( Line 583) Wittchen HU et al and Reference 43(Line 619) Kummerle S et al are not in english.

We thank the reviewer for noticing this. The references have been translated, with the note that the original sources are in German.

---

## [Decision Letter · Decision Letter 1]

16 Feb 2024

Patterns of social-affective responses to trauma exposure and their relation to psychopathology

PONE-D-23-22194R1

Dear Dr. Trautmann,

We’re pleased to inform you that your manuscript has been judged scientifically suitable for publication and will be formally accepted for publication once it meets all outstanding technical requirements.

Kind regards,

Lakshit Jain, MD

Academic Editor

PLOS ONE

Additional Editor Comments (optional):

Reviewers' comments:

Reviewer's Responses to Questions

**Comments to the Author**

1. If the authors have adequately addressed your comments raised in a previous round of review and you feel that this manuscript is now acceptable for publication, you may indicate that here to bypass the “Comments to the Author” section, enter your conflict of interest statement in the “Confidential to Editor” section, and submit your "Accept" recommendation.

Reviewer #1: All comments have been addressed

Reviewer #4: All comments have been addressed

2. Is the manuscript technically sound, and do the data support the conclusions?

Reviewer #1: Yes

Reviewer #4: Yes

3. Has the statistical analysis been performed appropriately and rigorously? 

Reviewer #1: Yes

Reviewer #4: I Don't Know

4. Have the authors made all data underlying the findings in their manuscript fully available?

Reviewer #1: Yes

Reviewer #4: Yes

5. Is the manuscript presented in an intelligible fashion and written in standard English?

Reviewer #1: Yes

Reviewer #4: Yes

6. Review Comments to the Author

Reviewer #1: The authors have thoughtfully and in a very detailed manner added adequate edits to the original manuscript to address the reviewer queries. They have also included the limitations of the study given the historical data availability. Though the data is old, the paper does add value to the factors that influence trauma symptoms and hence this manuscript would add to literature related to PTSD and I recommend acceptance of this article as it is presented in the revision.

Reviewer #4: (No Response)

7. PLOS authors have the option to publish the peer review history of their article (what does this mean?). If published, this will include your full peer review and any attached files.

Reviewer #1: No

Reviewer #4: No

---

## [Editor Report · Acceptance letter]

24 Feb 2024

PONE-D-23-22194R1 

PLOS ONE

Dear Dr. Trautmann, 

I'm pleased to inform you that your manuscript has been deemed suitable for publication in PLOS ONE. Congratulations! Your manuscript is now being handed over to our production team.

Kind regards, 

on behalf of

Dr. Lakshit Jain 

Academic Editor

PLOS ONE